# AdvNF: Reducing Mode Collapse in Conditional Normalising Flows using Adversarial Learning

**Vikas Kanaujia[1*], Mathias S. Scheurer[2†] and Vipul Arora[1‡]**

**1** Department Of Electrical Engineering, Indian Institute Of Technology Kanpur, India
**2** Institute for Theoretical Physics III, University of Stuttgart, Stuttgart, Germany

* kvikas@iitk.ac.in , † mathias.scheurer@itp3.uni-stuttgart.de , ‡ vipular@iitk.ac.in

## Abstract

Deep generative models complement Markov-chain-Monte-Carlo methods for efficiently sampling from high-dimensional distributions. Among these methods, explicit generators, such as Normalising Flows (NFs), in combination with the Metropolis Hastings algorithm have been extensively applied to get unbiased samples from target distributions. We systematically study central problems in conditional NFs, such as high variance, mode collapse and data efficiency. We propose adversarial training for NFs to ameliorate these problems. Experiments are conducted with low-dimensional synthetic datasets and XY spin models in two spatial dimensions.

# 1 Introduction

Many real-world problems require sampling from intractable multi-dimensional distributions. These samples could be useful for studying the behaviour of physical systems by estimating their statistical properties using Monte Carlo approximations. Sampling through such distributions has always been a challenge and is performed via perturbative approximations or Markov Chain Monte Carlo (MCMC) techniques [1]. In cases where variables are strongly coupled and there are no small parameters, perturbative approximations cannot be applied, and MCMC methods are used. To guarantee the asymptotic exactness of samples generated via MCMC methods, the Metropolis-Hastings algorithm (MH) is used, which makes use of model and target densities and can be applied even when these densities are only known up to a proportionality constant. However, MCMC techniques have their limitations, such as correlated sample generation, critical slowing down during phase transitions, and higher simulation costs.

In the past few years, several learning-based methods have been developed to sample from such distributions. Generative adversarial networks (GANs) [2–4] and variational autoencoders (VAEs) [5,6] have demonstrated remarkable efficacy in sampling distributions that are learned from given samples of the target distributions. VAEs are approximate density models as they provide approximate density values for the samples. GANs generate samples without explicitly estimating density values for samples; hence, they are also called implicit density models. Both of them do not guarantee the exactness of the samples. Furthermore, they cannot be modified or de-biased using methods like MH since they do not provide an exact model density. On the other hand, flow-based generative models such as Normalising flows (NF) [7,8] explicitly model the target distribution and provide exact model density values. They are used along with MH to guarantee the exactness of samples.

In physics applications, one is interested in sampling from probability distributions over physical configurations (e.g., the direction of each spin of a classical magnet) which are parameterized by a physical model. These physical models depend on a certain set of parameters, referred to as $c$ in the following, such as temperature $T$ or coupling constants. For example, in the Ising model and the XY model, the properties of the system depend on the ratio of temperature and the nearest-neighbour exchange (or, if included, further neighbour or ring-exchange) coupling constants. Varying these parameters can also drive the system through phase transitions, which has also been studied with machine-learning techniques [9–17]. One way to model such distributions is to train the generative model afresh for each setting of external parameters. For studying the properties of the system, samples are needed for several different settings of the external parameters. This leads to training the model repeatedly in different settings and, hence, increases the training cost. Many lattice theories have been

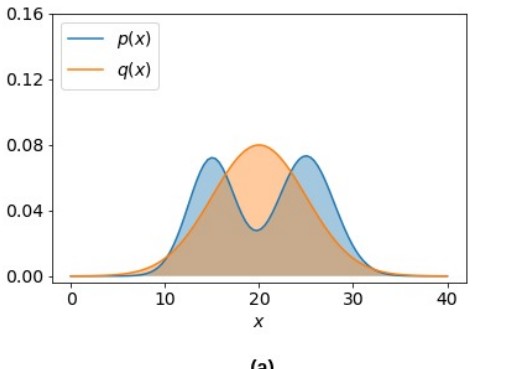 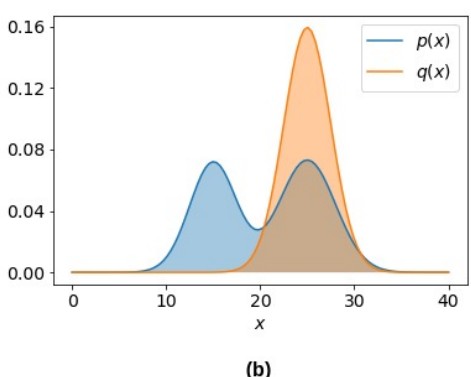

Figure 1: For the illustration of (a) mode-covering (FKL) and (b) mode-seeking behaviour (RKL), we show a comparison of toy density plots. Here $p(x)$ represents the univariate multi-modal target distribution and $q(x)$ represents the modeled distribution.

modelled in such a way using normalising flows [18–20]. The alternative way to model such distributions is to train the generative models conditioned on the external parameters. These external parameter-dependent distributions are commonly referred to as conditional distributions, where the set of external parameters is generally used to represent the condition. Many conditional generative models, such as conditional VAEs [21, 22], conditional GANs [23–26] and conditional NFs [27, 28] have been developed over time for sampling from such conditional distributions. Both approaches—repeated training and conditional modelling—have been the subject of substantial investigation over time. In the domain of lattice field theory, several generative models have been used. However, there are also certain shortcomings with these models. When the target distribution is multi-modal, the generative models may fail to model all the modes, leading to mode collapse. It becomes more prominent in the case of conditional models [29]. It could be defined as when the generator produces samples from only a few modes of the data distribution but misses several modes. In other words, the generator fails to generate data as diverse as the distribution of real-world data and rather covers only a few modes [30–32]. Several methods have been proposed to tackle this problem for GANs [30, 33]. Though NF models are known to suffer from mode collapse, this phenomenon has not been thoroughly investigated in these models. In particular, mode collapse has received very little research in the physical systems where these methods have been applied widely.

Before delving further into mode collapse, we must comprehend its causes and distinguish between the model's mode-covering and mode-seeking behaviours. Here we refer to $p(x)$ as the target distribution and $q(x)$ as the modelled distribution for mathematical representation. When the model is trained via forward KL divergence (FKL), i.e., $D_{KL}(p \parallel q) = \int p(x) ln(\frac{p(x)}{q(x)}) dx$. The model has mode-covering behaviour, which implies covering all the modes and, in addition, including other regions in the sample space where the distribution assigns a very low probability mass. When $p(x)$ is nonzero and $q(x)$ is near zero, FKL $\to \infty$. It penalises the model and brings $q(x)$ closer to $p(x)$. However, when $p(x)$ is multimodal and $q(x)$ is unimodal, optimal $q(x)$ tries to cover all the modes even if $q(x)$ is nonzero at places where $p(x)$ is near zero. It does not penalise such behaviour, which results in mode-covering behaviour in the model. On the other hand, when the model is trained via reverse KL divergence (RKL), i.e., $D_{KL}(q \parallel p) = \int q(x) \ln(\frac{q(x)}{p(x)}) dx$, the model has mode-seeking behaviour, which could be explained as follows: If $p(x)$ is near zero while $q(x)$ is nonzero, then RKL $\to \infty$ penalises and

forces $q(x)$ to be near zero. When $q(x)$ is near zero and $p(x)$ is nonzero, KL divergence would be low and thus does not penalise it. This causes $q$ to choose any mode when $p$ is multimodal, resulting in a concentration of probability density on that mode and ignoring the other high-density modes. Consequently, optimising RKL leads to a suboptimal solution, causing $q$ to have limited support, leading to mode collapse [34–36]. In a sense, mode collapse and mode seeking are equivalent because mode-seeking leads to focusing on a few modes only, which causes mode collapse. Figure 1 displays mode-covering and mode-seeking behaviours when the model is trained with forward KL and reverse KL, respectively, where the target distribution is univariate multimodal Gaussian.

NF models are trained via either forward or reverse KL divergence between the target $p$, and the modelled $q$ distribution. Reverse KL training causes mode collapse in NF models, as explained above. NF models trained via forward KL have been extensively applied to approximate Boltzmann distributions [27, 37–39]. These approaches require samples from the distribution, which are inefficiently obtained through MCMC simulations. Also, these models have mode-covering behaviour, resulting in a high variance of sample statistics [40, 41]. Other works, such as [42] in atomic solids, [41] in Alanine dipeptide, [43] in phi-4 theory, have used reverse KL to optimise NF models, but they suffer from mode collapse, resulting in biased sample statistics [43]. The problem of mode collapse in conditional NFs is not well studied. In this paper, we study the problem of mode collapse in conditional NFs. We also propose adversarial training of conditional NFs to overcome this problem. The advantages of our proposed method, AdvNF, could be summarised as follows: It minimises both the initial cost (by training from minimal number of MCMC samples) and the running cost (once trained, it can efficiently generate samples for any parameter) and improves the quality of the generated samples. Generally, MCMC methods have a high running cost (especially in critical regions), while other generative models have a higher initial cost. Sample generation through MCMC is a computationally time-intensive method. Besides that, MCMC simulation is fraught with many challenges. As it involves local updates, the generated samples are very much correlated, which further amplifies near the region of phase transition. The AdvNF model offers an alternate method to generate uncorrelated samples. Although training the AdvNF model is time-intensive in some ways, But it is a one-time investment; once the model is trained, any number of samples can be generated for any parameter value in a very negligible amount of time, even in critical regions. Moreover, in AdvNF (RKL), we further minimise the number of samples for training the model.

Over time, several generative models have been introduced for sample generation and applied across various domains. Among all those, normalising flows (NFs) have been extensively used in the physics domain because of their several features, like explicit density modelling and no requirement of samples for training the model (RKL), provided the Hamiltonian is known. This offers a unique advantage to NFs compared to other generation techniques. It greatly reduces the dependence on MCMC for generating training samples to learn the model. However, such learning also induces some problems. The model fails to learn the distribution completely. When distribution is multi-modal, the model is able to learn only a few modes, causing mode collapse. We have tried to provide a remedy for the above problem through our proposed model, AdvNF. This nudges the model learned through conditional NF (CNF) to learn all the modes by including adversarial learning in the training algorithm. While learning, some bias is introduced in the model as it was observed in observables. To reduce that bias, we use the Independent Metropolis-Hastings algorithm, where the trained model is used to generate samples, which are accepted or rejected based on acceptance probability. Here, we just refine the samples further without iterating the process repetitively. On the other hand, the performance of other generative models is heavily dependent on the amount of training data, which makes them costlier compared to our approach, where a minimal number of samples

are needed. Our main contributions are as follows:

- We show that normalising flows conditioned on some external parameters exhibit severe mode collapse when trained through reverse KL divergence, while those trained through forward KL divergence are found to be computationally inefficient. We study mode collapse on synthetic 2-D distributions (MOG-4, MOG-8, Rings-4), the XY model, and the extended XY model datasets. For synthetic datasets, mode collapse could be easily observed on 2-D sample plots; for other datasets, it may be observed through other evaluation metrics.

- We propose AdvNF, which uses adversarial training for NFs, and use it to model synthetic 2-D distributions, the XY model dataset and the extended XY model dataset. It gives a better approximation to the target distributions: As compared to reverse KL-trained NF, it substantially reduces mode collapse. When compared with forward KL-trained NF, it improves the observable statistics estimated through Monte Carlo from samples of the modelled distributions. The observable statistics obtained from AdvNF outperform implicit models such as GANs and approximate density models such as VAEs. When compared to conditional NFs, it performs better for most of the evaluation metrics.

- We use the Independent Metropolis-Hastings Algorithm to reduce bias in the modelled density as explained in Section 2.3.

- We show that our proposed method, AdvNF (RKL), yields almost identical results even when a very small ensemble size is chosen for training the model, hence reducing dependence on expensive MCMC simulations for data generation needed for training the model.

The rest of the paper is structured as follows: We briefly discuss the various generative modelling approaches that we investigate in this work in Section 2. In Section 3, we elaborate on our proposed method. Experimental details conducted on various datasets are presented in Section 4, along with descriptions of the various evaluation metrics chosen for comparing the performance. In Section 5, we compare the results obtained from our proposed model with various baselines. Lastly, Section 6 provides a summary and conclusion.

## 2   Generative Models

With the advancement in the field of deep neural networks, several architectures have been proposed that can model complex probability distributions effectively and generate new samples [44–48]. These models could be broadly classified into two categories [49]: explicit density models and Implicit density models. An explicit density model defines the density function of the distribution explicitly. For these models, density could either be computed exactly or approximately. For example, in Normalizing flows, density could be computed exactly, while VAEs allow for approximate computation. On the other hand, implicit density models generate samples directly without any tractable computation of model density. They indirectly interact with the model density during training, e.g., GAN [50] and GSN [51]. In this section, we will briefly explain Generative adversarial networks (GAN), Normalising flows (NF) and the Independent metropolis hastings algorithm (IMH), which will set the stage for the introduction of our proposed model in the next section. Readers familiar with these approaches can skip this section and directly proceed with Sec. 3.

## 2.1 Normalizing Flows

NFs model complex probability distributions via a series of simple bijective transformations on any known distribution from which samples could be easily generated [7,8]. A vector $z \in \mathbb{R}^N$, sampled from a known standard distribution $q_z(z)$, is transformed into $x \in \mathbb{R}^N$ via a chain of parametric bijective transformations.

$$x = T(z),\, T : \mathbb{R}^N \to \mathbb{R}^N \tag{1}$$

The density of $x$ is obtained by the change of variables formula,

$$q_x(x) = q_z(T^{-1}(x))|\det(J_{T^{-1}}(x))| \tag{2}$$

where $J_{T^{-1}}(x) = \partial T^{-1}/\partial x$ is the jacobian of $T^{-1}$. The parametric distribution $q_x(x)$ can be used to model a target distribution $p_x(x)$. The parameters of $T$ can be trained in two ways. If the samples from the target distribution are available, the model is trained by minimising the forward KL divergence (FKL) between the target distribution $p(x)$ and the modelled distribution $q(x)$, estimated as

$$KL(p(x) \,\|\, q(x)) = -\mathbb{E}_{p_x}[\log(q_z(T^{-1}(x))) + \log|\det J_{T^{-1}}(x)|] + const \tag{3}$$

On the other hand, if samples from the distribution are not available, the model is trained by minimising the reverse KL divergence (RKL) between the modelled distribution $q(x)$ and the target distribution $p(x)$, estimated as

$$KL(q(x) \,\|\, p(x)) = \mathbb{E}_{q_z}[\log(q_z(z)) - \log|\det J_T(z)| - \log p_x(T(z))] \tag{4}$$

To model conditional probability distributions, the parametric bijective transformation $T$, conditioned on an external parameter $c$, is expressed as $x = T(z; c)$.

## 2.2 Generative Adversarial Network (GAN)

GAN consists of two networks, the generator G, and the discriminator D. The generator is a mapping $G : z \to x$, which transforms sample $z \in \mathbb{R}^M \sim q_z(z)$ to generated data $x \in \mathbb{R}^N$ [49, 50]. The discriminator, $D : x \to (0,1)$ acts as a binary classifier, assigning a probability corresponding to each sample and quantifying whether the sample is a generated or true sample from the distribution. In general, generative models are trained by maximising the likelihood of samples from $p_x(x)$. Whereas in adversarial training, discriminator D is used to distinguish the modelled samples $x \sim q_x(x)$ from the target samples $x \sim p_x(x)$. It takes both actual data and generated data as input to the network. On the other hand, the generator tries to fool the discriminator by improving upon the generated samples so that the discriminator is unable to classify them as generated samples. Training proceeds by optimising the binary cross-entropy loss for the discriminator, while the generator G plays the role of an adversary that strives to maximise the loss. The loss function for the network is as follows:

$$V(G, D) = -\mathbb{E}_{x \sim p_x}[\log(D(x))] - \mathbb{E}_{z \sim q_z}[\log(1 - D(G(z)))] \tag{5}$$

## 2.3 Independent Metropolis-Hastings Algorithm (IMH)

Metropolis-Hastings algorithm belongs to the class of MCMC methods to generate samples from a probability distribution whose density function is either known exactly or known up to a certain proportionality constant [52, 53]. Sometimes it is difficult to sample from $p_x(x)$, but a Markov chain can be constructed to generate samples incrementally by sampling from

the proposal distribution $q(x'|x)$, provided the detailed balance principle is satisfied, which is given by

$$p(x')P(x', x) = p(x)P(x, x'). \tag{6}$$

Here, $P(x, x')$ is the transition probability from state $x$ to $x'$.

Metropolis-Hastings algorithm constructs a Markov chain asymptotically. It proposes a new sample $x'$ from the current sample $x$, and then stochastically accepts or rejects the sample with an acceptance probability given by

$$p_{\text{accept}}(x'|x) = \min\left(1, \frac{q(x|x')p(x')}{q(x'|x)p(x)}\right) \tag{7}$$

where $q(x'|x)$ is the probability of sample $x'$ given the previous sample x. In the independent metropolis sampler [54, 55], we draw sample $x'$ independent of previous sample x, i.e., $q(x'|x) = q(x')$ and the resulting expression for acceptance probability becomes

$$p_{\text{accept}}(x'|x) = \min\left(1, \frac{q(x)p(x')}{q(x')p(x)}\right). \tag{8}$$

## 3   Proposed Method

In this section, we describe the proposed adversarially trained conditional NFs. For $x \in \mathbb{R}^N$, let $p_x(x; c)$ be the target distribution conditioned on external parameter(s) $c$. The generative model with distribution $q_x(x; c)$ is implemented as an NF. We feed sample $z \in \mathbb{R}^N$ from a known distribution as input to the model, which is transformed by the neural network generator $T : \mathbb{R}^N \to \mathbb{R}^N$ to $x$. The model density can be written as

$$q_x(x; c) = q_z(T^{-1}(x; c))|\det(J_{T^{-1}}(x; c))| \tag{9}$$

The model can be trained so that $q_x(x; c)$ matches closely with $p_x(x; c)$. For adversarial training, another classifier network, $D : \mathbb{R}^N \to (0, 1)$ is defined. Due to exact density computation, various objective functions can be used to learn the model.

- The model can be trained by minimising the FKL divergence between the target distribution and the modelled distribution, i.e., $KL[p(x)||q(x)]$, provided the samples from the target distribution are available. The loss function is given as

$$L_{FKL}(T; c) = -\mathbb{E}_{p_x(x;c)}[\log(q_z(T^{-1}(x; c))) + \log|\det J_{T^{-1}}(x; c)|] \tag{10}$$

- The model can also be trained by minimising the RKL divergence between the modelled distribution and target distribution, i.e., $KL[q(x)||p(x)]$, provided the target distribution is known up to a certain proportionality. The loss function can be expressed as

$$L_{RKL}(T; c) = \mathbb{E}_{q_z(z)}[\log(q_z(z) - \log|\det J_T(z; c)| - \log(p_x(T(z; c)))] \tag{11}$$

- The model can also be trained adversarially by minimising the binary cross entropy (BCE) loss for $D$ and maximising the same for $T$. The loss is

$$L_{adv}(T, D; c) = -\mathbb{E}_{x \sim p_x}[\log(D(x; c))] - \mathbb{E}_{z \sim q_z}[\log(1 - D(T(z; c); c))] \tag{12}$$

The known limitations during training are that RKL loss does not cover all the modes, while FKL loss imports high variance in samples. Adding adversarial loss to one or both of these

improves the model performance. It allows the model to better explore and learn the unseen modes. It also helps the model reduce the sample variance. The final training objective can be written as follows:

$$O(T, D; c) = -\lambda_1 L_{adv}(T, D; c) + \lambda_2 L_{RKL}(T; c) + \lambda_3 L_{FKL}(T; c);$$  (13)

here, $\lambda_1, \lambda_2, \lambda_3 \in \mathbb{R}$ are hyperparameters. They are varied as per the schedule described in Algorithm 1 and Appendix C. Once the model gets trained, the NF generator can produce samples and also compute the exact probability density of each sample. Since the generator introduces bias in the samples due to approximation errors, we eliminate the bias by applying the independent Metropolis-Hastings algorithm. That is, samples are generated from the trained model distribution, which acts as the proposal distribution and is stochastically accepted or rejected based on acceptance probabilities. Fig. 2 shows the block diagram of our proposed model.

---

**Algorithm 1** Adversarial training of AdvNF
$\lambda_1, \lambda_2, \lambda_3$ are hyperparameters tuned during training

---

    Initialize flow, $q_x(x; c)$ parameterized by $T$
    Set $\lambda_1 = 0$,
    Set $\lambda_2, \lambda_3$ as per the experiment
    **while** $O(T, D; c)$ converges **do**
        Sample minibatch of m samples $z^{(1:m)}$ from $q_z(z)$
        Get minibatch of m samples $x^{(1:m)}$ from $p_x(x)$
        Compute $O(T, D; c)$
        Perform gradient descent on $O(T, D; c)$ to update parameters of $T$
        $T \leftarrow T - \eta \nabla_T(O(T, D; c))$
    **end while**
    Initialize discriminator parameterized by $D$
    Set $\lambda_1 > 0, \lambda_2, \lambda_3$ as per the experiment
    **for** iteration = 1 to K **do**
        Sample minibatch of m samples $z^{(1:m)}$ from $q_z(z)$
        Get minibatch of m samples $x^{(1:m)}$ from $p_x(x)$
        Compute $O(T, D; c)$
        Perform gradient descent on $O(T, D; c)$ to update parameters of $T$ and $D$
        $T \leftarrow T - \gamma \nabla_T(O(T, D; c))$
        $D \leftarrow D + \zeta \nabla_D(O(T, D; c))$
    **end for**

---

# 4  Experiments

In this section, we discuss the experiments conducted on various datasets, briefly describing them and the metrics chosen for analysis. Along with the XY model and the extended XY model datasets, we also conduct experiments on datasets where modes are readily observable on a 2-D sample plot and density can be explicitly expressed up to a certain proportionality constant

## 4.1  Synthetic 2-D datasets

Under the synthetic datasets, we use a 2-dimensional mixture of Gaussians (MOG-4, MOG-8) and concentric rings (Rings-4). On synthetic datasets, mode collapse can be easily observed

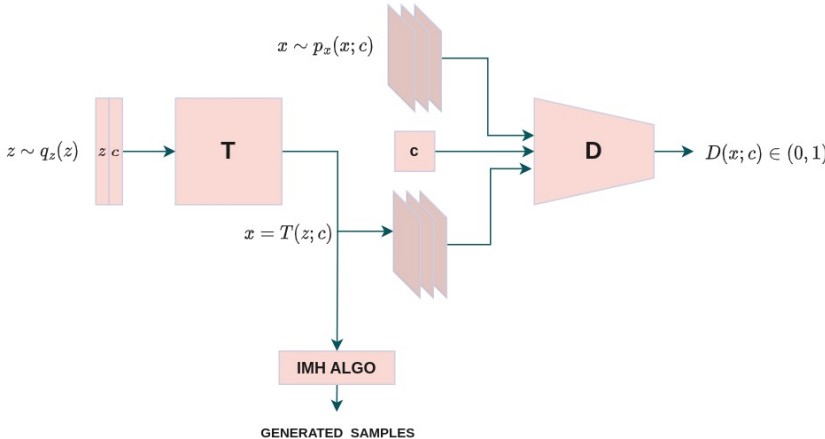

Figure 2: Schematics of AdvNF (Conditional Flow-Adversarial Model).

since the true distribution and its modes are known. We generate these datasets by sampling from the Gaussian mixture model. The probability density function for a mixture of Gaussians can be analytically written as

$$p(x) = \sum_{i=1}^{N} a_i \mathcal{N}(x; \mu_i, \Sigma_i), \tag{14}$$

where $\mathcal{N}(x; \mu_i, \Sigma_i)$ represents gaussian distribution with mean $\mu_i \in \mathbb{R}^2$ and $\Sigma_i \in \mathbb{R}^{2X2}$ as covariance matrix. Here $a_i > 0$, is the weight of the $i^{th}$ Gaussian component, and N refers to the number of Gaussian components. The probability density function for the concentric rings dataset is given by

$$p(r, \theta) = \sum_{i=1}^{N} a_i \mathcal{N}(r; r_i, \sigma_i^2) \mathcal{U}(\theta; 0, 2\pi) \tag{15}$$

Here, the distribution of each ring is represented by the product of two distributions, namely the Gaussian distribution represented by $\mathcal{N}(r; r_i, \sigma_i^2)$, where $r_i$, $\sigma_i$ denote mean radius and standard deviation, and the uniform distribution represented by $\mathcal{U}(\theta; 0, 2\pi)$, with support set $(0, 2\pi)$. While $a_i$ refers to the weight of each ring and N corresponds to the number of rings.

We take the following methods as baselines: (1) CNF-MH (FKL): The conditional NF model is trained by minimising FKL and then applying the IMH algorithm to accept or reject the samples generated from the model. (2) CNF-MH (RKL): The conditional NF model is trained by minimising RKL, and then the generated samples are accepted or rejected using IMH. (3) CNF-MH (FKL+RKL): The conditional NF model is trained by minimising both FKL and RKL and then applying IMH.

The generators in AdvNF and all other CNF models are implemented by using affine coupling layers [56]. In MOG, the model is conditioned on the mean of the Gaussian component, whereas for Rings-4, the radius of the rings is used to condition the model. Further details on the architectures and hyperparameters used in the algorithm are provided in the Appendices. For training and testing the model, 4000 samples have been generated separately. In MOG-4 and Rings-4, there are 1000 samples for each Gaussian component, while in MOG-8, 500 samples for each Gaussian component are used for training the model.

## 4.2   XY Model and Extended XY Model dataset

Although the method proposed in this work can in principle be applied to any physical model, we have chosen the XY model and its extended version to verify and validate our proposed

method. The XY model [9] is a statistical mechanics model where the spin at each site $i$ of a two-dimensional lattice is described by a two-component unit vector $\boldsymbol{s}_i = (\cos\theta_i, \sin\theta_i)$, $\theta_i \in [0, 2\pi)$. The total energy is given by

$$E_{XY}(\theta) = -J \sum_{\langle i,j \rangle} (\boldsymbol{s}_i \cdot \boldsymbol{s}_j) = -J \sum_{\langle i,j \rangle} \cos(\theta_i - \theta_j), \tag{16}$$

where $\langle i, j \rangle$ denotes nearest neighbors, $J \in \mathbb{R}$ is coupling constant and $\theta = \{\theta_i\}$ is a shorthand for the spin configuration. For concreteness, we here focus on $N \times N$ square lattices. To demonstrate that the success of our approach is not due to any peculiar properties of this model, we also consider an extended version of the usual XY model, where we add ring-exchange interactions,

$$E_{EXY}(\theta) = -J \sum_{\langle i,j \rangle} \cos(\theta_i - \theta_j) - K \sum_{(i,j,k,l) \in \square} \cos(\theta_i - \theta_j + \theta_k - \theta_l). \tag{17}$$

Here, $\sum_{(i,j,k,l) \in \square}$ is the sum over all elementary plaquettes of the square lattice with $i, j, k, l$ at its corners. By elementary plaquettes, we imply the smallest 4-site square clusters of spins appearing on the lattice. The second term ($\propto K$) in Eq. 17, which is often referred to as ring exchange, has full square-lattice symmetries and spin-rotation invariance, just as the usual XY model.

In both cases, the probability of a spin configuration $\theta$, for given $c = J/T$ or $c = (J/T, K/T)$, is just given by the respective Boltzmann factor with proper normalization; for instance, for the extended XY model, it reads as

$$P(\theta; J/T, K/T) = \frac{1}{Z(J/T, K/T)} e^{-\frac{E_{EXY}(\theta)}{T}}, \quad Z(J/T, K/T) = \sum_\theta e^{-\frac{E_{EXY}(\theta)}{T}}, \tag{18}$$

where we set the Boltzmann constant to unity. This determines any observable quantity such as mean energy and mean magnetization. For instance, the former is

$$\langle E \rangle = \sum_\theta P(\theta; J/T, K/T) E_{EXY}(\theta). \tag{19}$$

We use $8 \times 8$ and $16 \times 16$ size square lattices for training the model. We generate the training dataset by using the MH algorithm for 32 values of temperature $T$, evenly spaced in the range of $[0.05, 2.05]$ for the XY model dataset, setting $J$ to unity. A total of 320,000 samples have been generated, with 10,000 corresponding to each temperature. We use a uniform distribution as proposal distribution $q_z(z)$. Around 30,000 initial samples have been discarded from the MCMC chain on account of burn-in or thermalization. For the extended XY model dataset, we generate 500,000 samples for 50 values of temperature, evenly spaced in the range of $[0.50, 3.50]$, with 10,000 samples against each temperature by the MH algorithm with both $J$ and $K$ set to unity. In addition, we also train the model on samples of lattice size $32 \times 32$ for 10 values of temperature T, evenly spaced in the range of $[0.85, 1.25]$ for the XY model dataset at $J = 1$. We use the MH algorithm to generate training dataset for the same, with 10,000 samples against each temperature. To have a minimum correlation among samples in the training data, a configuration is added to the set after every 320 MCMC steps for an $8 \times 8$ lattice, 1280 steps for a $16 \times 16$ lattice and 5120 steps for a $32 \times 32$ lattice .

Incorporating an inductive bias to match the topology of the data helps with better modelling. Hence, the data is transformed to manipulate the support of the density function. Generally, NFs are easier to learn in Euclidean spaces, while the XY-model spin configuration data lies on a circular topology. For this reason, we project this circular manifold to $\mathbb{R}^N$ before applying the RNVP architecture [56] to model flows. Once trained, we project $\mathbb{R}^N$ space back to the circular manifold. We use the following types of projections:

- Tan Transformation (tan):
  Since the spin at each lattice site is represented by angle an $\theta \in [0, 2\pi)$, to transform circular space to Euclidean space, we use a projection $x : [0, 2\pi) \rightarrow \mathbb{R}$ defined as:

$$x(\theta) = \tan((\alpha + (1 - 2\alpha)\frac{\theta}{4})) \tag{20}$$

  Here $\alpha$ is a small regularization parameter, to reduce the effect of the boundary; we choose $\alpha = 10^{-4}$ in all experiments.

- Sigmoid Transformation ($\sigma$):
  Here, we use the logit function to project $\theta$ to Euclidean space. It is defined as:

$$x(\theta) = \log(\frac{(\alpha + (1 - 2\alpha)\frac{\theta}{2\pi})}{1 - (\alpha + (1 - 2\alpha)\frac{\theta}{2\pi})}) \tag{21}$$

  Both projections are invertible and are taken into account while computing likelihood.

We train our proposed model, AdvNF, conditioned on temperature ($T$) to generate samples. We use 5000 samples for training, 1000 samples for validation, and 1000 samples for test evaluation against each temperature value. We use the same training procedure for training all the variants of AdvNF.

We compare our model with the following networks as baselines: (1) CGAN [23]: Conditional GAN model to generate samples, conditioned on temperature T in the XY model dataset. (2) C-HG-VAE [57]: it is an approximate density model applying VAE, conditioned on temperature T, to model the distribution. It minimises standard Evidence lower bound (ELBO) loss [5] along with an additional term computing the square of the difference between the energy of the ground truth and the generated sample, acting as a regularizer. (3) Implicit GAN [26]: It is also a conditional GAN model. It optimises adversarial loss along with a regularizer, which minimises output bias in the sample, and another term contributed by an additional auxiliary network trained to maximise the output entropy of the system. (4) CNF-MH: Here, we train the conditional NF model to generate samples and subsequently apply IMH to de-bias the samples.

## 4.3   Evaluation Metrics

To evaluate model performance, we compute certain metrics common to all datasets, like Negative log likelihood (NLL) and Acceptance rate (AR). While for the XY model dataset, to assess the efficacy of our model, besides NLL and AR, we have also chosen some specific evaluation metrics that quantify the extent to which the ensemble of generated samples follows the true distribution, like Percent overlap and Earth mover distance (EMD). Thermodynamic observables calculated using MCMC-generated data are used for comparison and metric evaluations in the XY model and extended XY model datasets. To this end, we focus on mean magnetization and mean energy. The distribution of these observables obtained from generated samples is compared with the distribution computed from data obtained through MCMC simulations in these metrics. We briefly explain these metrics in the following.

### 4.3.1   Negative Log Likelihood (NLL)

NLL estimates how efficiently the model fits the data. Mathematically, it is expressed as

$$\text{NLL} = -\mathbb{E}_{p_x}[\log(q(x; \theta))], \tag{22}$$

where $x$ represents the samples from the true distribution $p_x(x)$, while $q_x(x)$ is the modelled distribution. Mode collapse in the model could be effectively estimated through NLL computation. A low NLL value reflects the closeness of the modelled distribution to the true distribution, while a higher value implies the model does not capture all the modes of the true distribution effectively.

### 4.3.2  Percent overlap (%OL)

Percent overlap measures the similarity between two distributions by computing overlap between their corresponding histograms, where both the histograms are normalised to unit sum. Mathematically, it is expressed as

$$\%\text{OL} = \sum_i \min(p_x(i), p_y(i)); \tag{23}$$

here, $p_x$ and $p_y$ are the distributions, and $i$ corresponds to the bin index. For the histogram of magnetization, we employ 40 bins in the $[0,1]$ range, and for the energy, we use 80 bins in the $[-2,0]$ range.

### 4.3.3  Earth Mover Distance (EMD)

It also measures the similarity between two distributions by calculating the least amount of work needed to turn one pile of distribution into another pile of distribution, where the distribution has been represented as a pile of dirt and work is quantified as the product of the amount of dirt moved and the distance by which it is moved.

$$W(p_x, p_y) = \sum_{j=-\infty}^{\infty} \sum_{k=-\infty}^{j} |(p_x(k) - p_y(k))| \tag{24}$$

### 4.3.4  Acceptance Rate (AR)

It refers to the ratio of the number of samples accepted to the total number of samples evaluated for the detailed balance principle in the MH algorithm.

$$\text{AR}(\%) = \frac{N_{\text{accepted}}}{N_{\text{accepted}} + N_{\text{rejected}}} \times 100 \tag{25}$$

where $N_{\text{accepted}}$ and $N_{\text{rejected}}$ represent the number of samples accepted and rejected, respectively, following the detailed balance principle as explained in Sec. 2.3.

## 5  Results

### 5.1  MOG-4, MOG-8 and Rings-4

For synthetic datasets, we compute NLL and Acceptance Rate for comparison. In Table 1, it can be observed that our proposed model AdvNF improves upon the NLL against each CNF-MH variant across all the datasets. The most striking improvement could be observed when comparing CNF-MH(RKL) with AdvNF(RKL). This is because Conditional Normalising Flow trained via RKL i.e. CNF-MH(RKL) suffers from heavy mode collapse, thereby leading to high NLL value. In MOG-4 and MOG-8 dataset distributions, the CNF-MH(RKL) model does not capture all the modes. Two modes are missing for the MOG-4 dataset, and one for the MOG-8 dataset, as shown clearly in the sample plot in Fig. 3, whereas for the Rings-4 distribution,

Table 1: Results for synthetic datasets (MOG-4, MOG-8 and Rings-4)

| | MOG-4 | | MOG-8 | | Rings-4 | |
|---|---|---|---|---|---|---|
| | NLL↓ | AR↑ | NLL↓ | AR↑ | NLL↓ | AR↑ |
| CNF-MH | | | | | | |
| FKL | 0.80 | 91.75 | 0.81 | 81.98 | 2.06 | 43.02 |
| RKL | 46.82 | 95.63 | 10.21 | 57.15 | 2274.07 | 90.57 |
| FKL & RKL | 0.79 | 94.15 | 0.84 | 57.09 | 1.88 | 78.90 |
| AdvNF | | | | | | |
| FKL | 0.79 | 94.16 | 0.80 | 88.93 | 2.02 | 46.43 |
| RKL | 0.78 | 96.95 | 0.79 | 85.79 | 1.99 | 91.58 |
| FKL & RKL | 0.78 | 96.11 | 0.80 | 75.68 | 1.84 | 82.76 |

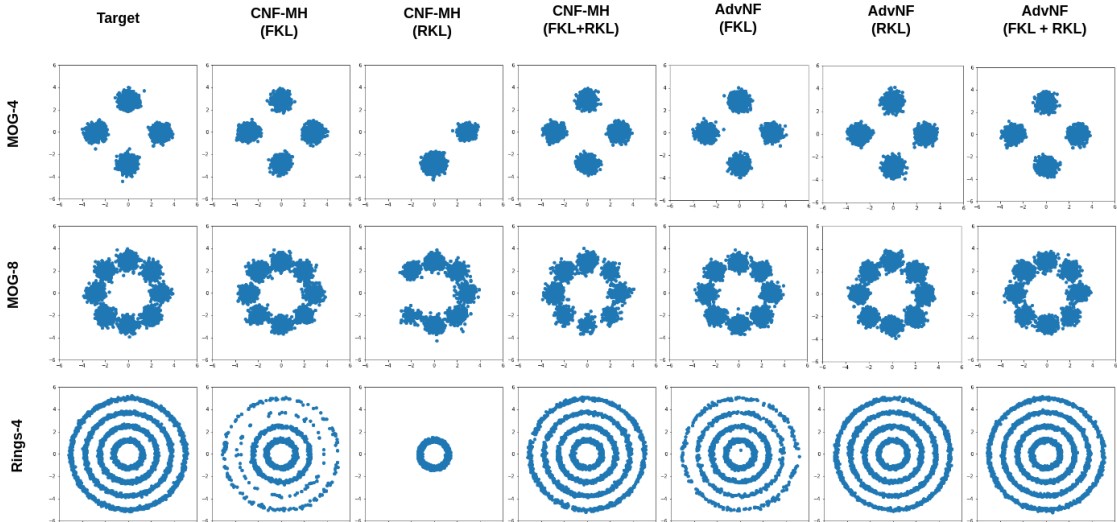

Figure 3: Sample plot for MOG-4, MOG-8 and Rings-4 distribution drawn by generating samples from AdvNF and CNF-MH variants. Mode collapse can be observed on all synthetic datasets for CNF-MH(RKL) variants

mode collapse further aggravates. Almost three rings are completely missing; the model focuses only on a single mode (the innermost ring) and completely fails to generate data from the outermost three rings. On the other hand, our proposed model, AdvNF has been able to capture all modes among all distributions, which could be seen in the sample plot. This improvement is also reflected in NLL as well. There is almost 1100 times improvement in it for the Rings-4 dataset, 60 times for MOG-4 and 12 times for the MOG-8 dataset. In Fig. 4, we show how the adversarial loss term pulls the RKL-trained model out of mode collapse. We initially keep the adversarial loss term higher by setting $\lambda_1$ high; it provides a sudden jolt to the network that brings it out of mode collapse and then gradually reduces it as the training progresses. It can be seen that as training advances, the model gradually learns to cover all the modes.

The conditional normalising flow (CNF) model, when trained via FKL, has mode-covering behaviour. However several samples are rejected on application of the IMH algorithm, resulting in a very sparsely populated sample plot. This behaviour could be prominently observed in the Rings-4 dataset in Fig. 3, where the acceptance rate is low (43%). Similar behaviour is also observed when the model is jointly trained via both FKL and RKL. However, the accep-

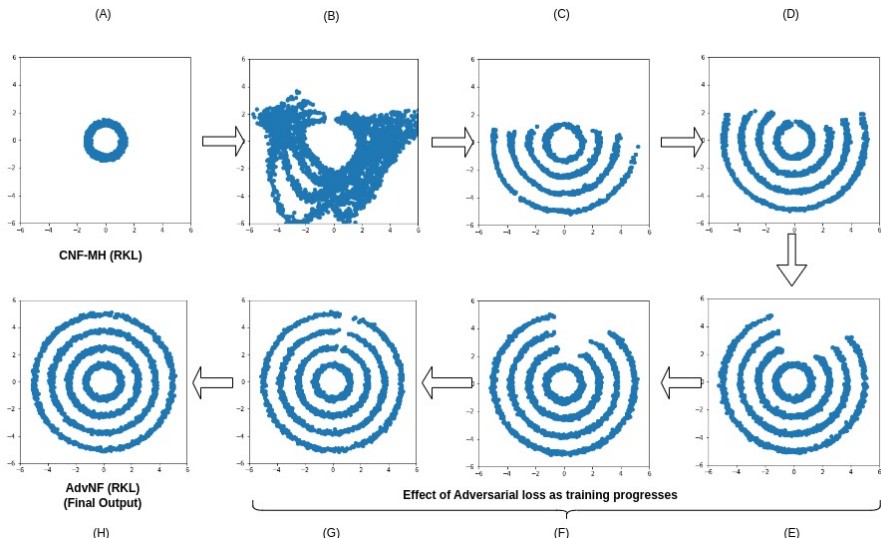

Figure 4: Sample plots for the Rings-4 distribution highlight the effect of adversarial loss as training progresses and illustrate how it comes out of mode collapse and converges to the desired target distribution. (A) the model distribution (trained through CNF-MH (RKL)) has collapsed to a few modes; (B) shows the effect of adding adversarial loss with a high adversarial loss weight $\lambda_1$. (C)-(G) show the model distribution gradually converging to the target distribution as $\lambda_1$ is decreased with epochs. (H) shows the sample plot when the model AdvNF (RKL) has been fully trained or converged to the target distribution.

tance rate is much better than CNF trained via FKL only. Our model's variants AdvNF(FKL) and AdvNF(FKL & RKL) corresponding to CNF-MH variants, improve marginally upon the NLL, but it leads to a higher acceptance rate comparatively. The main reason behind the slight improvement in NLL could be attributed to the mode-covering aspect of FKL training. Since most of the modes are already covered, therefore very marginal improvement in NLL when trained adversarily. From the results, it could be established that the variants of our proposed model, AdvNF, outperform the corresponding CNF-MH variants. In addition, AdvNF (RKL) performs better than AdvNF (FKL) and AdvNF (FKL & RKL) among its several variations.

## 5.2   XY Model dataset

To compare our proposed model with the baselines as explained in Sec. 4.2, we compute mean magnetization and mean energy as observables, which are both functions of temperature. Tables 2 & 3 compare the results between AdvNF and the various baselines implemented for the XY model dataset and the extended XY model dataset, respectively, for the lattice configurations of size $16 \times 16$. Results for lattice size of $8 \times 8$ and $32 \times 32$ are presented under Additional results in Appendix A. It quantifies %OL and EMD by taking MCMC samples as ground truth. It can be observed that mean magnetization decreases and energy increases with temperature for all models trained on any lattice size. Though CGAN and CHG-VAE follow similar behaviour, there are inherent biases and larger variances in the observable statistics, which could be inferred from Tables 2 & 3 as well as visualised in Fig. 5. The implicit GAN model, to some extent, tries to reduce bias in the samples and improve upon the observable characteristics. But it still has lower performance compared to our proposed model. While in AdvNF and CNF-MH, the inherent bias in the samples is mostly reduced due to the application of the IMH. Fig. 5 shows that our approach, AdvNF, and the ground truth (MCMC simulations) coincide

Table 2: Results for the XY model dataset(16 × 16 lattice size) at setting $J = 1$. Evaluation metrics, as defined in Sec. 4.3, are computed along with the standard deviation over 1000 configurations and averaged across all temperatures.

| | NLL↓ | AR↑ | Energy | | Magnetization | |
| | | | % OL↑ | EMD↓($10^{-3}$) | % OL↑ | EMD↓($10^{-3}$) |
|---|---|---|---|---|---|---|
| CGAN | - | - | 13.3 ± 22.1 | 134 ± 89 | 4.7 ± 8.4 | 267 ± 84 |
| C-HG-VAE | - | - | 7.1 ± 4.9 | 169 ± 37 | 30.6 ± 16.9 | 176 ± 58 |
| Implicit GAN | - | - | 66.0 ± 21.4 | 19 ± 19 | 62.7 ± 14.0 | 41 ± 25 |
| CNF-MH | | | | | | |
| FKL | 272.6 | 3.6 | 57.4 ± 20.2 | 15 ± 9 | 54.7 ± 20.9 | 33 ± 26 |
| RKL | 312.8 | 23.4 | 80.2 ± 10.7 | 7 ± 5 | 80.1 ± 14.1 | 17 ± 20 |
| FKL & RKL | 270.0 | 8.3 | 68.1 ± 17.7 | 12 ± 11 | 69.6 ± 20.5 | 27 ± 34 |
| AdvNF | | | | | | |
| FKL | 272.1 | 3.7 | 60.9 ± 21.1 | 11 ± 8 | 51.5 ± 22.3 | 39 ± 36 |
| RKL | 290.7 | 21.5 | 83.7 ± 9.9 | 4 ± 3 | 79.3 ± 12.4 | 15 ± 15 |
| FKL & RKL | 269.8 | 8.8 | 76.6 ± 12.0 | 5 ± 3 | 70.6 ± 15.0 | 20 ± 17 |

quite well when compared to other baselines.

In comparison to the CGAN, C-HG-VAE, and implicit GAN models, our proposed model, AdvNF, yields the best performance across all metrics and lattice sizes. On the other hand, when compared to CNF-MH variants, our model has better results over most of the metrics for the XY model and the extended XY model datasets. Furthermore, among its variants, AdvNF (RKL) outperforms AdvNF (FKL) and AdvNF (FKL & RKL) in terms of observable statistics and acceptance rate, while it has a comparable NLL with that of the other two variants, suggesting a reduction in mode collapse. Analogous to the analysis for synthetic datasets, NLL results are consistent in the XY model and the extended XY model datasets as well. The NLL metric has fairly improved on all AdvNF variants compared to CNF-MH variants. This improvement is specifically significant for AdvNF (RKL) compared to CNF-MH (RKL), where mode collapse is supposed to be severe and evident as well through the NLL values. While for the remaining of the AdvNF variants, the improvement is marginal compared to the respective CNF-MH variants, which could be accounted for by the FKL loss in their objective function, which has mode-covering behaviour, causing inherent improvement in the NLL metric.

All AdvNF variants, with the exception of the RKL-trained form, have seen a slight improvement in acceptance rate when compared to the corresponding CNF-MH variants. The reason for the higher acceptance rate in CNF-MH (RKL) could be attributed to sampling from only a few modes by the model CNF-MH (RKL), resulting in higher acceptance of samples by IMH. While the AdvNF (RKL) model seeks to cover all modes by transitioning from one mode to another during IMH, this transitioning often results in the rejection of slightly more samples when the model traverses or jumps from one mode to another, thereby reducing the acceptance rate. This behaviour has not been observed on synthetic datasets, which could be attributed to the low dimensionality of the datasets. In a low-dimensional space, traversal from one mode to another does not cause a large shift in the probability measure. However, a similar shift in a higher dimension may result in a greater change in the probability measure, which leads to the rejection of samples during the application of independent MH.

The CNF model trained via RKL does not require samples of the data distribution. Samples from the base distribution are enough to train the model, which is usually chosen to be multivariate Gaussian. Even though models trained via RKL do not require samples from the

Table 3: Results for the XY Extended Model dataset ($16 \times 16$ lattice size) at setting $K/J = 1$. Evaluation metrics, along with standard deviation are computed over 1000 configurations, averaged across all temperatures.

| | NLL↓ | AR↑ | Energy | | Magnetization | |
|---|---|---|---|---|---|---|
| | | | % OL↑ | EMD↓($10^{-3}$) | % OL↑ | EMD↓($10^{-3}$) |
| CGAN | - | - | 19.2 ± 19.3 | 254 ± 174 | 31.3 ± 25.5 | 204 ± 146 |
| C-HG-VAE | - | - | 37.7 ± 27.8 | 85 ± 45 | 20.9 ± 22.8 | 236 ± 121 |
| Implicit GAN | - | - | 45.4 ± 14.6 | 55 ± 26 | 79.2 ± 13.6 | 33 ± 39 |
| CNF-MH | | | | | | |
|   FKL | 334.0 | 3.9 | 52.5 ± 16.8 | 17 ± 13 | 52.0 ± 19.6 | 38 ± 35 |
|   RKL | 384.3 | 16.0 | 67.4 ± 19.1 | 14 ± 15 | 66.4 ± 19.5 | 38 ± 46 |
|   FKL & RKL | 332.2 | 9.4 | 65.1 ± 19.2 | 13 ± 9 | 65.6 ± 16.7 | 26 ± 22 |
| AdvNF | | | | | | |
|   FKL | 333.6 | 4.1 | 54.2 ± 17.7 | 16 ± 11 | 53.4 ± 18.2 | 35 ± 22 |
|   RKL | 363.1 | 13.3 | 67.9 ± 17.8 | 12 ± 11 | 67.6 ± 19.6 | 30 ± 38 |
|   FKL & RKL | 334.1 | 8.0 | 66.8 ± 15.3 | 11 ± 7 | 66.1 ± 17.3 | 27 ± 23 |

target distribution, adversarial learning on top of it introduces the need for true samples of the distribution. In adversarial training, the discriminator network needs to be fed by actual samples of the target distribution as well as generated samples from the generator. Therefore, our proposed model, AdvNF, which is trained adversarially, needs true samples of the distribution. Even the RKL variant of AdvNF requires samples from the distribution for training. Generating samples via MCMC or HMC for a high-dimensional distribution is time-inefficient and hence costlier. Besides, these methods also introduce correlation among the samples due to the Markov property. In general, the model's performance varies depending on the ensemble size or number of training samples when the model is trained using FKL or simultaneously trained using FKL and RKL. To study the effect of ensemble size on our model, we train all variants of AdvNF with different ensemble sizes. We chose 4 ensembles of size 100, 512, 1024 and 5120 samples for each temperature to train the model. The performance evaluation can be seen in Table 4. It can be inferred from the table that AdvNF(FKL) and AdvNF(FKL & RKL) variants are completely dependent on the ensemble size. The larger the ensemble size, the higher the performance. Reducing ensemble size degrades the observable statistics. While for AdvNF(RKL), observable statistics are almost similar in value. Besides that, mode collapse is also reduced irrespective of the ensemble size, which could be substantiated by almost similar NLL values.

As discussed in Sec. 4.2, we project the data into Euclidean space either through the use of the tan or $\sigma$ transform as a preprocessing step, before feeding the data into the model. In another experiment, we study the outcomes of using these transforms with our model, AdvNF, as shown in Table 5. We can see that compared to the tan transformation, the results from $\sigma$ transformation are relatively better.

# 6  Conclusion

In this work, we propose an adversarily trained conditional normalising flow model, AdvNF, for generating samples of physical models which allows to avoid the problem of mode collapse. We

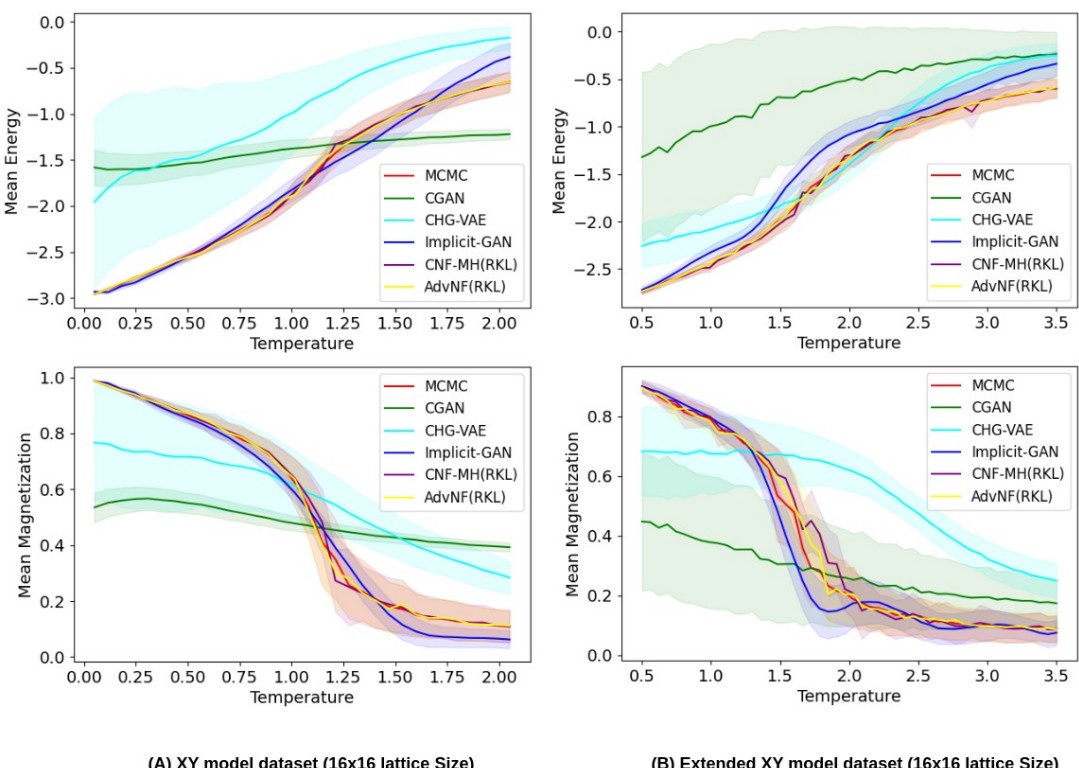

**(A) XY model dataset (16x16 lattice Size)**      **(B) Extended XY model dataset (16x16 lattice Size)**

Figure 5: Comparison plot of observables (mean energy and mean magnetization) for AdvNF and various other baseline models referred in Sec. 4.2, with MCMC acting as ground truth. The line represents the mean value, and the shaded area represents the standard deviation. 10000 samples are generated at each temperature to compute observables for all the models. (A) XY model dataset ($16 \times 16$ lattice size) at setting $J = 1$. (B) Extended XY model dataset ($16 \times 16$ lattice size) at setting $K/J = 1$.

focus on NF models that have been extensively used to generate samples in lattice field theory [18–20, 58–60]. Most of the works use RKL based approach for training these flow models. The main reason for the popularity of NFs could be attributed to RKL divergence loss, where the model does not require samples from the true distribution to train it. Samples from the base distribution are sufficient to train the model, provided samples from the base distribution could be easily generated and the true distribution could be mathematically represented even if it is up to a certain proportionality constant. The first criteria can be readily met by using the multivariate Gaussian as the base distribution, which is easy to sample from. The second condition can be satisfied by using datasets or models that use Boltzmann distributions. This is one of the key reasons these models are chosen: producing the samples in a critical region using MCMC techniques is still quite challenging. As we approach closer to the critical region, critical slowing down has a significant impact on sample generation. Furthermore, samples generated via MCMC methods are always correlated to a certain extent.

However, there has not been a systematic study of NFs in the physics domain. When trained via RKL, these NF models have been found to have a severe mode collapse problem, which deteriorates even further when the model is trained conditionally. Meanwhile, training via FKL leads to high variance in sample statistics. Here, we study the mode collapse comprehensively on several synthetic datasets as well as XY model datasets and illustrate how conditional NF

Table 4: Performance Comparison of our model (AdvNF) trained on different sample size for the XY model dataset (8 × 8 lattice size) at setting $J = 1$. Evaluation metrics, along with standard deviation are computed over 1000 configurations, averaged across all temperatures.

| Sample Size | NLL↓ | AR↑ | Energy % OL↑ | Energy EMD↓($10^{-3}$) | Magnetization % OL↑ | Magnetization EMD↓($10^{-3}$) |
|---|---|---|---|---|---|---|
| AdvNF (FKL) | | | | | | |
| 100 | 79.2 | 5.7 | 64.6 ± 18.3 | 17 ± 16 | 64.9 ± 18.2 | 32 ± 28 |
| 512 | 74.5 | 5.4 | 61.3 ± 17.5 | 14 ± 13 | 62.2 ± 17.3 | 34 ± 30 |
| 1024 | 72.0 | 10.4 | 70.1 ± 18.6 | 11 ± 9 | 70.6 ± 15.4 | 27 ± 24 |
| 5120 | 71.3 | 11.2 | 74.3 ± 13.7 | 9 ± 9 | 73.2 ± 13.8 | 25 ± 29 |
| AdvNF (RKL) | | | | | | |
| 100 | 74.7 | 29.6 | 82.1 ± 11.6 | 7 ± 7 | 81.4 ± 12.3 | 18 ± 20 |
| 512 | 73.1 | 26.3 | 80.7 ± 11.0 | 7 ± 6 | 80.3 ± 11.8 | 18 ± 19 |
| 1024 | 73.6 | 24.8 | 80.7 ± 10.9 | 8 ± 7 | 80.7 ± 10.9 | 18 ± 17 |
| 5120 | 73.7 | 35.9 | 84.9 ± 9.7 | 6 ± 5 | 84.0 ± 10.5 | 17 ± 19 |
| AdvNF (FKL & RKL) | | | | | | |
| 100 | 77.9 | 10.4 | 68.9 ± 13.9 | 13 ± 12 | 68.2 ± 16.7 | 32 ± 30 |
| 512 | 73.1 | 11.0 | 72.3 ± 14.4 | 11 ± 10 | 70.0 ± 15.5 | 28 ± 29 |
| 1024 | 71.0 | 17.1 | 73.9 ± 9.8 | 9 ± 5 | 73.7 ± 12.4 | 21 ± 16 |
| 5120 | 68.0 | 31.6 | 82.8 ± 13.3 | 6 ± 5 | 82.9 ± 14.0 | 14 ± 13 |

models are unable to emulate the true distributions. To address the aforementioned problem, we introduce an adversarial training approach in our proposed model AdvNF, which proves to be effective in reducing mode collapse and generating quality samples from the model, as validated through several experiments on the XY model and the extended XY model datasets. We introduce three variants of AdvNF, namely FKL, RKL, and a combination of both FKL and RKL.

AdvNF always needs true samples of the distribution for adversarial training, where the discriminator always needs samples of the true distribution to differentiate between generated samples and true samples. Nevertheless, AdvNF can be trained with very few true samples. Our experiments confirm that the RKL variant of AdvNF can be trained with extremely small ensemble sizes, and can still minimise mode collapse and preserve comparable sample statistics. Generating such a small ensemble size is generally feasible with MCMC methods, say.

In order to validate the efficacy of the proposed method, we experimented on the XY model dataset and the extended XY model dataset for various lattice sizes. We compared our method with several baselines, which can be broadly classified into three categories: GAN-based methods (CGAN, Implicit GAN), VAE-based methods (C-HG-VAE), and CNF-based variants. We observed that our method shows improved results compared to all the baselines. In addition to that, with an increase in lattice size, the results with the proposed method, AdvNF, improved compared to all the baselines. However, as compared to GAN-based and VAE-based approaches, the relative improvement increases with increasing lattice size but remains essentially constant when compared to CNF-based variations.

Overall, it can be concluded that AdvNF offers a good substitute for flow-based methods. It is an interesting and important open question for future work to evaluate how this approach performs on other more complex classical models and whether it yields a similar boost in performance for sampling for quantum systems. Besides that, the baselines as well as the

Table 5: Comparison among tan and $\sigma$ variants of AdvNF.

| | NLL↓ | AR↑ | Energy % OL↑ | Energy EMD↓$(10^{-3})$ | Magnetization % OL↑ | Magnetization EMD↓$(10^{-3})$ |
|---|---|---|---|---|---|---|
| FKL($\sigma$) | 71.3 | 11.2 | 74.3 ± 13.7 | 9 ± 9 | 73.2 ± 13.8 | 25 ± 29 |
| RKL($\sigma$) | 73.7 | 35.9 | 84.9 ± 9.7 | 6 ± 5 | 84.0 ± 10.5 | 17 ± 19 |
| FKL & RKL($\sigma$) | 68.0 | 31.6 | 82.8 ± 13.3 | 6 ± 5 | 82.9 ± 14.0 | 14 ± 13 |
| FKL(tan) | 72.2 | 12.6 | 71.9 ± 15.4 | 9 ± 7 | 72.3 ± 18.0 | 27 ± 36 |
| RKL(tan) | 82.4 | 20.0 | 73.2 ± 15.7 | 10 ± 8 | 74.8 ± 16.6 | 22 ± 22 |
| FKL&RKL(tan) | 68.6 | 22.2 | 80.1 ± 7.7 | 7 ± 5 | 80.9 ± 8.1 | 17 ± 15 |

proposed method model the joint pdf in a non-factorizable way. Hence, with an increase in lattice size, the model size and training time increase. This presents a future direction to work towards factorizable models, which can be efficiently scaled up to any lattice size.

## Acknowledgements

This work is supported by Core Research Grant of SERB, Govt. of India (project no. 003466).

## A  Additional Results

Table 6: Results for the XY model dataset ($8 \times 8$ lattice size) at setting $J = 1$. Evaluation metrics, along with standard deviation are computed over 1000 configurations, averaged across all temperatures.

| | NLL↓ | AR↑ | Energy % OL↑ | Energy EMD↓$(10^{-3})$ | Magnetization % OL↑ | Magnetization EMD↓$(10^{-3})$ |
|---|---|---|---|---|---|---|
| CGAN | - | - | 14.5 ± 8.1 | 303 ± 70 | 49.3 ± 23.9 | 159 ± 78 |
| C-HG-VAE | - | - | 30.8 ± 5.4 | 221 ± 50 | 73.4 ± 10.1 | 87 ± 45 |
| Implicit GAN | - | - | 76.5 ± 12.6 | 34 ± 29 | 79.3 ± 11.9 | 31 ± 26 |
| CNF-MH | | | | | | |
| FKL | 72.8 | 11.4 | 65.5 ± 16.4 | 16 ± 13 | 69.5 ± 17.8 | 27 ± 26 |
| RKL | 93.3 | 45.0 | 82.3 ± 8.5 | 9 ± 7 | 84.2 ± 8.9 | 16 ± 15 |
| FKL & RKL | 68.0 | 25.9 | 75.9 ± 11.1 | 11 ± 10 | 79.3 ± 13.4 | 18 ± 19 |
| AdvNF | | | | | | |
| FKL | 71.3 | 11.2 | 74.3 ± 13.7 | 9 ± 9 | 73.2 ± 13.8 | 25 ± 29 |
| RKL | 73.7 | 35.9 | 84.9 ± 9.7 | 6 ± 5 | 84.0 ± 10.5 | 17 ± 19 |
| FKL & RKL | 68.0 | 31.6 | 82.8 ± 13.3 | 6 ± 5 | 82.9 ± 14.0 | 14 ± 13 |

In this section, we present the results for $8 \times 8$ lattice size for both the XY model and the extended XY model dataset in Tables 6 and 7. In Fig. 6, we show the comparison plot of observables for various baselines as mentioned in Sec. 4.2 and AdvNF. Besides that, we also present the results for $32 \times 32$ lattice size trained for XY model dataset at $J = 1$ in Table 9 along

with comparison plot of observables for various baselines in Fig. 7. Improvement in evaluation metrics can be seen for AdvNF compared to the various baselines. This further substantiates the effectiveness of the proposed method even for larger systems.

Table 7: Results for the XY Extended Model dataset ($8 \times 8$ lattice size) at setting $K/J = 1$. Evaluation metrics, along with standard deviation are computed over 1000 configurations, averaged across all temperatures.

|  | NLL↓ | AR↑ | Energy | | Magnetization | |
|---|---|---|---|---|---|---|
|  |  |  | % OL↑ | EMD↓($10^{-3}$) | % OL↑ | EMD↓($10^{-3}$) |
| CGAN | - | - | $26.3 \pm 14.3$ | $369 \pm 139$ | $57.3 \pm 28.3$ | $170 \pm 156$ |
| C-HG-VAE | - | - | $23.1 \pm 7.3$ | $254 \pm 56$ | $69.7 \pm 8.0$ | $93 \pm 36$ |
| Implicit GAN | - | - | $68.8 \pm 5.1$ | $56 \pm 25$ | $73.5 \pm 8.9$ | $43 \pm 23$ |
| CNF-MH |  |  |  |  |  |  |
|   FKL | 87.1 | 13.2 | $66.6 \pm 11.8$ | $17 \pm 11$ | $70.3 \pm 10.5$ | $29 \pm 22$ |
|   RKL | 94.7 | 39.5 | $78.2 \pm 12.0$ | $14 \pm 17$ | $80.2 \pm 12.1$ | $23 \pm 30$ |
|   FKL & RKL | 83.7 | 26.1 | $73.8 \pm 10.6$ | $13 \pm 10$ | $76.4 \pm 9.7$ | $20 \pm 19$ |
| AdvNF |  |  |  |  |  |  |
|   FKL | 85.0 | 14.1 | $66.4 \pm 10.4$ | $16 \pm 11$ | $70.7 \pm 11.8$ | $24 \pm 16$ |
|   RKL | 87.4 | 34.3 | $78.8 \pm 8.3$ | $12 \pm 9$ | $80.2 \pm 7.9$ | $19 \pm 15$ |
|   FKL & RKL | 83.4 | 27.1 | $74.5 \pm 11.4$ | $13 \pm 8$ | $77.3 \pm 10.3$ | $24 \pm 21$ |

## B   Model Details

In this section, we provide details about the implementation of the model. For synthetic datasets, 10 conditional affine coupling layers [56] are used to implement both CNF and AdvNF. Each coupling layer consists of two dense layers with 32 neurons each and ReLU as activation. The output consists of two layers, one for scaling and one for translation. In the scaling layer, 1 neuron is used with Tanh activation, while the translation layer consists of 1 neuron with linear activation. The mean of the corresponding Gaussian in the MOG dataset and the radius of the concentric ring in the Rings-4 dataset are provided as conditioning inputs to the model. Training is performed with a batch size of 256 using the Adam optimizer with an initial learning rate of $1 \times 10^{-4}$, which is subsequently decayed by applying piecewise constant decay as a rate scheduler. The discriminator model in AdvNF is implemented by a simple neural network. It is composed of four dense layers with 64 neurons each, a fifth dense layer with an additional 8 neurons, and a final layer with an output of just one neuron. ReLU has been applied as an activation in all layers, except the final output layer. The generator model is the same as the CNF implementation. The discriminator is also trained using the Adam optimizer with an initial learning rate of $5 \times 10^{-5}$, which decays in the same manner as the generator's learning rate decays.

For the XY model and the extended XY model dataset, we use conditional affine coupling layers to implement the CNF model. The detailed architecture for the conditional Affine coupling layer can be seen in Fig. 8. Input to layer is an $N \times N \times 1$ matrix representing spin configuration at each lattice site, pre-processed by either using $\sigma$ or $tan$ transformation, where $N = \{8, 16, 32\}$. Temperature is given as a conditional input by repeating it to create a matrix of the same shape as that of the input. Two conv layers are used with 64 filters of size $3 \times 3$

Table 8: Results for the XY Model dataset (32×32 lattice size) at setting $J = 1$. Evaluation metrics, along with standard deviation are computed over 1000 configurations, averaged across all temperatures.

| | NLL↓ | AR↑ | Energy % OL↑ | Energy EMD↓($10^{-3}$) | Magnetization % OL↑ | Magnetization EMD↓($10^{-3}$) |
|---|---|---|---|---|---|---|
| CGAN | - | - | 2.3 ± 3.1 | 123 ± 34 | 23.5 ± 16.8 | 162 ± 71 |
| C-HG-VAE | - | - | 22.6 ± 19.2 | 67 ± 34 | 17.5 ± 19.7 | 259 ± 147 |
| Implicit GAN | - | - | 57.4 ± 24.2 | 16 ± 10 | 14.0 ± 9.6 | 169 ± 63 |
| CNF-MH | | | | | | |
|   FKL | 1251 | 0.9 | 19.3 ± 24.1 | 42 ± 26 | 15.0 ± 18.8 | 209 ± 131 |
|   RKL | 1363 | 7.2 | 71.2 ± 20.3 | 7 ± 6 | 38.7 ± 28.2 | 114 ± 87 |
|   FKL & RKL | 1248 | 1.8 | 60.0 ± 12.8 | 8 ± 3 | 44.0 ± 15.3 | 44 ± 21 |
| AdvNF | | | | | | |
|   FKL | 1247 | 0.8 | 24.7 ± 24.3 | 30 ± 21 | 13.4 ± 12.3 | 163 ± 117 |
|   RKL | 1255 | 2.4 | 72.5 ± 10.0 | 5 ± 2 | 47.7 ± 12.3 | 52 ± 39 |
|   FKL & RKL | 1235 | 2.0 | 70.7 ± 8.9 | 6 ± 2 | 46.2 ± 12.8 | 48 ± 33 |

with periodic padding and relu as activation for $N = \{8, 16\}$. For $N = 32$, we use 256 filters of size $3 \times 3$ for both the conv layers. While output consists of two layers, one for scaling factor $s$ and the other for translation, $t$. Each layer consists of one filter of size $3 \times 3$. The scaling layer applies tanh activation, while the translation layer applies linear activation. The generator model for AdvNF remains the same as that used for the CNF model. For experiments with the XY model dataset, we have used 24 affine coupling layers for a lattice size of $8 \times 8$, 50 coupling layers for a lattice size of $16 \times 16$, and 24 coupling layers for a lattice size of $32 \times 32$. Whereas, for the extended XY model dataset, we have used 30 conditional affine coupling layers for $8 \times 8$ size and 50 layers for $16 \times 16$ size, keeping the coupling layer architecture similar to the one used in the XY model dataset. Discriminator networks differ with different lattice sizes. Architecture for lattice size $8 \times 8$ is shown in Fig. 9. For higher lattice sizes of $16 \times 16$ and $32 \times 32$, architecture remains similar with an increase in the number of conv and maxpooling layers.

## C Hyperparameter Details

When the CNF model is trained, hyperparameters $\lambda_2$ and $\lambda_3$ are set as per the objective function used. $\lambda_1$ corresponding to adversarial loss, is set to 0 for all CNF variants. When FKL is minimised, $\lambda_3$ is set to 1.0, and the rest of the hyperparameters are set to 0. During RKL minimization, $\lambda_2$ is set to 1.0, and the rest of the hyperparameters are set to 0. When the model is jointly trained by minimising both FKL and RKL, we set both $\lambda_2$ and $\lambda_3$ to some finite value. We chose two sets of hyperparameters. In the first set, both $\lambda_2$ and $\lambda_3$ are set to 1, while in the second set, $\lambda_2$ was set to 0.5 and $\lambda_3$ was set to 1. The second set gave better performance for both synthetic datasets and the XY model dataset and has been reported in the paper.

For training the model adversarially in AdvNF, we first train the model by minimising the objective function until convergence is reached, setting the hyperparameters the same as used in CNF training. After that, we include adversarial loss in the training objective and train the model for certain epochs, depending on the dataset. Hyperparameter $\lambda_1$ corresponds to ad-

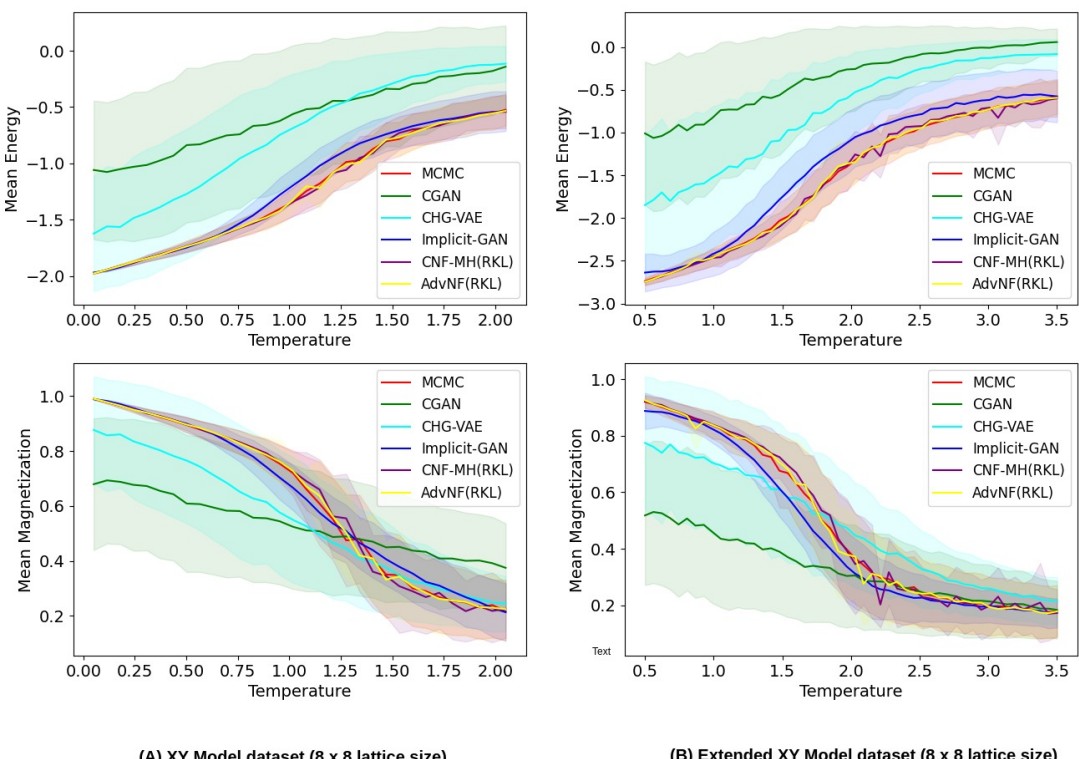

**(A) XY Model dataset (8 x 8 lattice size)**          **(B) Extended XY Model dataset (8 x 8 lattice size)**

Figure 6: Comparison plot of observables (mean energy and mean magnetization) for AdvNF and various other baseline models, with MCMC acting as ground truth. The line represents the mean value, and the shaded area represents the standard deviation. 10000 samples are generated at each temperature to compute observables for all the models. (A) XY model dataset ($8 \times 8$ lattice size) at setting $J = 1$. (B) Extended XY model dataset ($8 \times 8$ lattice size) at setting $K/J = 1$.

Table 9: Hyperparameter settings for training AdvNF.

|            | Synthetic Datasets | | | XY model Dataset | | | Extended XY model | | |
|------------|-------------|-------------|-------------|-------------|-------------|-------------|-------------|-------------|-------------|
|            | $\lambda_1$ | $\lambda_2$ | $\lambda_3$ | $\lambda_1$ | $\lambda_2$ | $\lambda_3$ | $\lambda_1$ | $\lambda_2$ | $\lambda_3$ |
| FKL        | 0.0 | 0.0 | 1.0 | 0.0 | 0.0 | 1.0 | 0.0 | 0.0 | 1.0 |
| RKL        | 0.0 | 1.0 | 0.0 | 0.0 | 1.0 | 0.0 | 0.0 | 1.0 | 0.0 |
| FKL & RKL  | 0.0 | 0.5 | 1.0 | 0.0 | 0.5 | 1.0 | 0.0 | 1.0 | 1.0 |
| AdvNF      |     |     |     |     |     |     |     |     |     |
| FKL        | 1.0 | 0.0 | 1.0 | 100.0 | 0.0 | 1.0 | 100.0 | 0.0 | 1.0 |
| RKL        | 1.0 | 1.0 | 0.0 | 10.0 | 1.0 | 0.0 | 5.0 | 1.0 | 0.0 |
| FKL & RKL  | 1.0 | 0.5 | 1.0 | 1.0 | 0.5 | 1.0 | 1.0 | 1.0 | 1.0 |

versarial loss. Its final value is chosen through hyperparameter tuning by monitoring the NLL value. However, its initial value is set based on RKL and FKL losses. Generally, adversarial loss hovers around 0.6 to 2.0, irrespective of any dataset, while RKL and FKL losses are comparatively higher. For synthetic datasets, both of these RKL and FKL losses vary around 0.5 to 4.0.

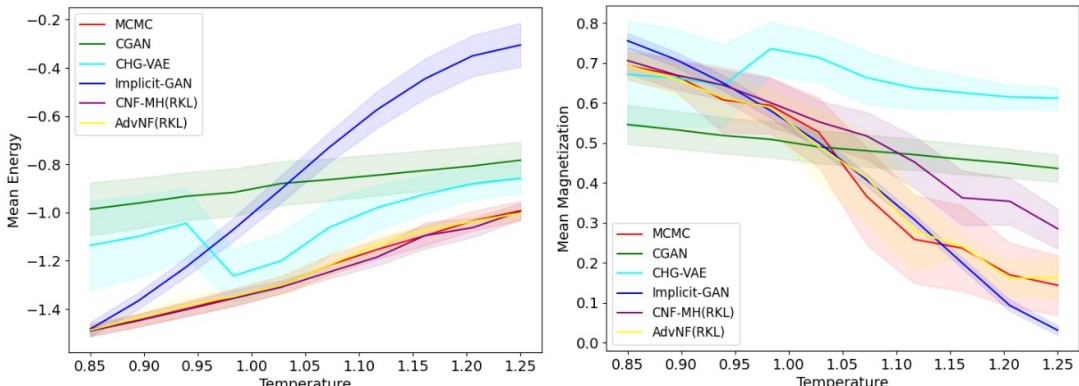

Figure 7: Comparison plot of observables (mean energy and mean magnetization) for AdvNF and various other baseline models, with MCMC acting as ground truth for XY model dataset ($32 \times 32$ lattice) at setting $J = 1.10000$ samples are generated at each temperature to compute observables for all the models.

Initially, we set the value of $\lambda_1$ to be high for these datasets based on their RKL and FKL losses. The final value of $\lambda_1$ is set to 1.0, obtained through hyperparameter tuning. For the XY Model dataset, in AdvNF (RKL), we initially set $\lambda_1$ to a high value (for instance, 100 for 8x8, set on the basis of RKL loss) so as to provide an initial jolt, which makes the model come out of the few modes and traverse other modes of distribution as well, and then gradually reduce its value as the training progresses based on the NLL value, which leads to the final value of $\lambda_1$ being set to 10.0. Reducing $\lambda_1$ further deteriorates the performance comparatively as gradients from RKL loss start dominating the gradients from adversarial loss. A similar process of hyperparameter tuning has been followed in RKL + FKL as well, resulting in the final value of $\lambda_1$ being set to 1.0 based on improvement in NLL. While in FKL, due to its mode-covering behaviour and to make adversarial loss dominate over FKL loss, the initial value is generally kept high. The final value has also been found to be on the higher side. The final values of the hyperparameter used for training could be referred to from Table 8. We train the model using a batch size of 256 samples to optimise the generator and discriminator using Adam as an optimizer. For the generator, the initial learning rate is set to $5 \times 10^{-5}$, while for the discriminator, the learning rate is set to $5 \times 10^{-5}$, which is subsequently decayed by applying piecewise constant decay as a rate scheduler. In Figs. 5,6 and 7, mean energy and mean magnetization for the XY model dataset and the extended XY model dataset have been plotted against temperature for samples generated from different models and compared with the reference MCMC-generated samples. Through these plots, it can be visually inferred that there is maximum overlap of observables (mean energy and mean magnetization) for the AdvNF. The exact %OL has been reported in Tables 2,3,6,7 & 8 in the paper.

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

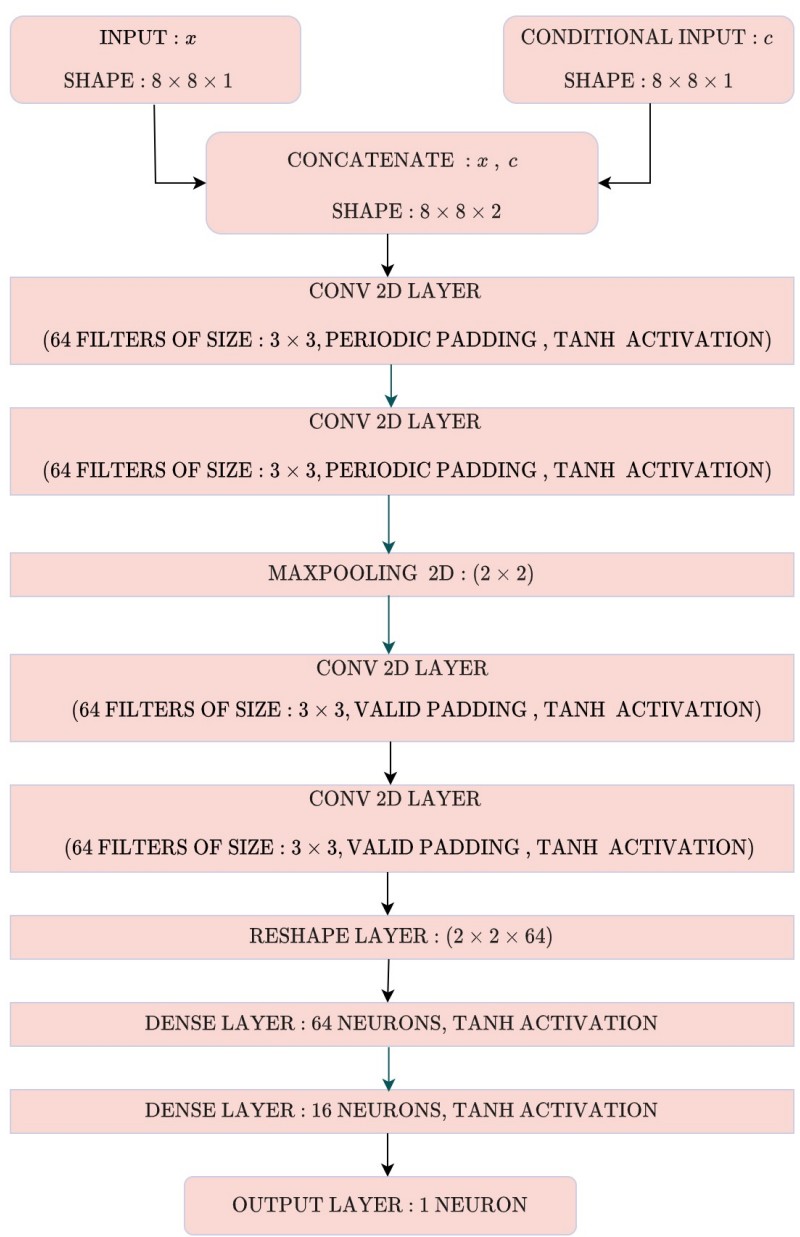

Figure 9: Architecture of Discriminator for XY-Model dataset.