# Peer review of "AdvNF: Reducing Mode Collapse in Conditional Normalising Flows using Adversarial Learning"

_SciPost Physics_

## Round 1 · Referee Report · Anonymous (Referee 1) · 2024-3-13

Report
n the manuscript entitled "Reducing Mode Collapse in Conditional Normalising Flows (NFs) using Adversarial Learning" the authors propose the use of adversarial training to address these issues. They conduct experiments on synthetic datasets and XY spin models to demonstrate the effectiveness of their approach. The paper highlights the importance of deep generative models in efficiently sampling from high-dimensional distributions and compares with other methods. The study focuses on improving the sampling efficiency and accuracy of NFs through adversarial learning techniques. The work aims to enhance the performance of NFs in modeling complex probability distributions, particularly in physics applications where sampling from conditional distributions is crucial.
The main focus of the paper is on addressing the challenges in conditional Normalizing Flows , such as mode collapse, high variance, and data efficiency. The authors utilizes adversarial training to mitigate mode collapse in NFs conditioned on external parameters. They conduct experiments on synthetic datasets and XY spin models to demonstrate the effectiveness of their approach. The paper compares with conditional NF models trained through reverse KL divergence and forward KL divergence, showing that Adversarial training significantly reduces mode collapse and improves the accuracy of observable statistics estimated through Monte Carlo simulations. The study aims to enhance the performance of NFs in modeling complex probability distributions, particularly focusing on sampling from conditional distributions efficiently.
The paper is well-written and clear, tackling an engaging subject. It includes detailed descriptions of the procedures. I believe the paper is worthy of publication once the authors address the following comments:
- To compare the proposed model with the baselines the authors compute mean magnetization and mean energy as observables as functions of temperature for the XY model, for a lattice of size 16 × 16. Although the results seems to be correct. I wonder why the authors uses a system so small. Syntethic data for XY model can be generated for larger systems.
- It would be interesting if the authors would discuss the effect of finite size on adversarial models compared to the baseline.
Author: Vipul Arora on 2024-04-11 [id 4410]
(in reply to Report 1 on 2024-03-13)
Thanks to the reviewer for carefully reading the manuscript and giving valuable comments.
The referee writes:
1.To compare the proposed model with the baselines the authors compute mean magnetization and mean energy as observables as functions of temperature for the XY model, for a lattice of size 16 × 16. Although the results seems to be correct. I wonder why the authors uses a system so small. Syntethic data for XY model can be generated for larger systems.
** Our response:**
In this paper, we have proposed a theory to address mode-collapse in NF models trained via RKL, which have been prominently applied in several Physics model. Therefore to validate the theory, we did experiments first on synthetic datasets and then on XY model dataset for 8x8 and 16x16 lattice sizes. However, the model can be scaled up to larger lattices as well in the similar way as proposed in the paper. We have added the results for 32x32 lattice size trained on 10 temperature settings in the revised manuscript and got improvement in results compared to the existing baselines similar to that seen for smaller lattices. This further substantiates the effectiveness of our method even for larger systems. Please refer Table 8 and Fig. 7 in the revised manuscript.
Another point to note here is that the baselines as well as the proposed method model the joint pdf in a non-factorizable way. Hence, with increase in lattice size, the model size and training time increase. For future work, we intend to work towards factorizable models, which could be efficiently scaled up to any lattice size [ https://doi.org/10.48550/arXiv.2308.08615].
We have updated the above point for future work in Conclusion in revised manuscript. Please refer last paragraph under Conclusion, Page-19.
| Model | NLL | AR(%) | % OL (Energy) |EMD (Energy)|% OL (Mag.) | EMD (Mag.) | |--------------- |:-----:|:-----:|:------------:|:----------:|:----------:|:----------:| | CGAN | - | - | 2.3 +/-3.1 | 123 +/- 34 | 23.5 +/- 16.8 | 162 +/- 71 | | C-HG-VAE | - | - | 22.6 +/- 19.2 | 67 +/- 34 | 17.5 +/- 19.7 | 259 +/- 147 | | Implicit GAN | - | - | 57.4 +/- 24.2 | 16 +/- 10 | 14.0 +/- 9.6 | 169 +/- 63 | | CNF (FKL) | 1251 | 0.9 | 19.3 +/- 24.1 | 42 +/- 26 | 15.0 +/- 18.8 | 209 +/- 131 | | CNF (RKL) | 1363 | 7.2 | 71.2 +/- 20.3 | 7 +/- 6 | 38.7 +/- 28.2 | 114 +/- 87 | | CNF (FKL & RKL) | 1248 | 1.8 | 60.0 +/- 12.8 | 8 +/- 3 | 44.0 +/- 15.3 | 44 +/- 21 | | AdvNF (FKL) | 1247 | 0.8 | 24.7 +/- 24.3 | 30 +/- 21 | 13.4 +/- 12.3 | 163 +/- 117 | | AdvNF (RKL) | 1255 | 2.4 | 72.5 +/- 10.0 | 5 +/- 2 | 47.7 +/- 12.3 | 52 +/- 39 | | AdvNF (FKL & RKL) | 1235 | 2.0 | 70.7 +/- 8.9 | 6 +/- 2 | 46.2 +/- 12.8 | 48 +/- 33 |
The referee writes:
2.It would be interesting if the authors would discuss the effect of finite size on adversarial models compared to the baseline.
** Our response:**
In this work, we have validated our proposed method on XY model dataset for lattice sizes of 8x8 , 16x16 and 32x32. We have compared our method with several baselines, which could be broadly classified into 3 categories: GAN based methods (CGAN , Implicit GAN), VAE based method (C-HG-VAE) and CNF based variants. We observed that our method shows improved results compared to all the baselines. In addition to that, with an increase in lattice size, the results with the proposed method, AdvNF, improve compared to all the baselines. However, as compared to GAN-based and VAE-based approaches, the relative improvement increases with increasing lattice size but remains essentially constant when compared to CNF-based variations.
We have added the above discussion in revised manuscript in Conclusion, Paragraph 4, Page no. 18, line no 12.
Anonymous on 2024-04-12 [id 4412]
(in reply to Vipul Arora on 2024-04-11 [id 4410])
Results of 32x32 lattice put in the previous reply have not been rendered properly. Please refer to Table 8 in the revised manuscript.
Author: Vipul Arora on 2024-04-11 [id 4409]
(in reply to Report 2 on 2024-03-24)The referee writes:
Our response:
The referee writes:
Our response:
The referee writes:
Our response:
The referee writes:
Our response:
The referee writes:
Our response:
The referee writes:
Our response:
The referee writes:
Our response:
The referee writes:
Our response:
The referee writes:
Our response:
The referee writes:
Our response:

---

## Round 1 · Referee Report · Anonymous (Referee 2) · 2024-3-24

Strengths
-
The manuscript is clearly written.
-
The proposed method really improves the problems of the previous methods, for example, the mode collapse.
-
Many results of different methods are compared.
-
Many metrics are used to evaluate the advantages and disadvantages of different methods.
Weaknesses
-
There lack some explanations of the results in the main text or in the captions: the parameters and the sample sizes etc..
-
Some important curves are not plotted, for example, the energy and the magnetisation curves of the CNF results.
-
The choosing of lamba_1, 2,3 is not well explained.
Report
Referee Report:
The manuscript entitled " AdvNFL: Reducing Mode Collapse in Conditional Normalising Flows using Adversarial Learning" by Kanaujia et. al. studies a learning process of Markov-chain-Monte-Carlo sampling via the learning process which combines the normalised flow-based (NF) learning with the adversarial learning(ADV).
The authors claim that in lattice field theory the RKL(Reverse KL divergence)-based CNF approaches without the adversarial learning are used to generate samples because no true Monte-Carlo data are required. However, the training suffers from mode collapse. On the other hand, CNF with forward KL divergence (FKL) has mode-seeking behavior, and it requires more true MC data to learn. In order to overcome the mode collapse and mode seeking behavior, they propose AdvNFL, where a discriminator is added together with a CNF to train the sampling. Several metrics are proposed to test if the proposed learning process is improved compared to the other methods: Negative likelyhood (NLL) which should be decreased, percent overlap (LO%), which should be increased, Earth Mover Distance (EMD), which should be decreased, and Acceptance Rate (AR) evaluated by the independent Metropolis-Hasting algorithm, which should be increased, if the modelled distribution is close to the true distribution. They found that with AdvNFL method the learning results solve the mode collapse problem and fit the energy and magnetisation curves calculated by the MCMC method.
This manuscript is clearly written with a lot of experiments and details. However, there are some issues that should be clarified:
1) A common raised question for the machine learning process applied in physics is: what do you gain by using the proposed method? For example, in the proposed AdvNFL method, true MCMC or IMH sampling data have to be used, and for a generative model, the learning process is normally heavy, therefore what is the gain compared to the physics process without ML?
On the other hand, for some models, where no simple and efficient global update method is known, th MCMC could be difficult and time consuming to be calculated, people proposed the machine to train an effective Hamiltonian to simplified the Monte-Carlo methods (PRB, 95, 041101 (R) 2017). If the authors want to use their method to such problems without known global updates, the calculations and training will become difficult. Therefore compared to these old methods, what do the authors gain by the new method?
2) Adding a discriminator is like to add a JS divergence to the NF model. The JS divergence is KL(p||(p+q)/2) + KL(q||(p+q)/2), which looks similar to RKL + FKL process with some suitable lambda_3 and lambda_2. Could the authors explain why a discriminator should be added to the CNF model to improve the learning instead of only using RKL + FKL?
3) A question related to the previous one: In Table 8, the choosing of lambda_1, lambda_2 and lambda_3 are listed. For the synthetic datasets in the AdvNF, lambda_1 is always chosen to be 1. However, in the XY, lambda_1 sis chosen as 100(FKL), 10(RKL), and 1 (RKL + FKL), while in the extended XY model, lambda_1 = 100 (FKL), and 5 (RKL) and 1(RKL + FKL). Why are they chosen like that?
4) In the manuscript, both MCMC and IMH are used. MCMC is used to generate the energy and magnetisation curves to compare with other ML results. However, in the main text, there is no explanation of which kind of true data are produced to be fed into the discriminator? Is that IMH or MCMC?
5) In Fig.5 and Fig. 6, the energy and magnetisation curves using different methods are plotted. It is true that the proposed AdvNFL method can improve the precision of the energy and magnetisation. However, for the magnetisation of the extended XY model, there are several instable regions that the magnetisation suddenly increases or decreases with sharp peaks. What is the reason? Could the authors elaborate that?
6) Related to 5): The author claims that the CNF model trained via RKL is interesting since it does not require the true data samples due the RKL divergence. Therefore it should be interesting to see the energy and magnetisation curves of the CNF (RKL) model and compare them with the ones by AdvNFL. However, in Fig.5 and Fig.6, they are not included.
7) In Fig.5 and 6 the caption is badly written and the informations are scarse. For example, what are the model parameter for the XY model and the extended XY model? For the XY model we can set J=1, however, for the extended XY model one needs K/J to describe the model. Another question is what is the sample size to calculate all results by MCMC, CGAN, CHG-VAE, implicit GAN and AdvNFL (RKL)? They are also not included in the main text.
8) Related to 7): It is well known that machine learning is a statistical model, therefore the results are statistical. For Fig.5 and Fig6, the curves are calculated by one-shot training or are mean values of many-shot trainings? If the former, the authors should do the many-shot trainings more than 100 or 200 times to obtain the mean values, while if the latter, the authors should clarify the training times.
9) Actually the algorithm of GAN has some stability issue about finding the arg min max log(P(G,D). First of all, it should fix G and find the maximal of D, than change G to find the min of max log(P). However, when G changed, the maximum of D could also be changed, which causes the instability of a discriminator. This also happens in AdvNFL since a discriminator is also added to the CFL. Could this kind of instability affects the results?
10) There are some typos in the text, for example, above the Eq.(7): given by given by. The authors should check the manuscript more carefully.
The manuscript deserves to be published in SciPost if the questions raised above are properly answered and the manuscript is well revised.

---

## Round 2 · Referee Report · Anonymous · 2024-4-19

Report

The authors have addressed all comments adequately. I deem the manuscript suitable for publication.

Recommendation

Publish (meets expectations and criteria for this Journal)

---

## Round 2 · Referee Report · Anonymous · 2024-4-29

Report

The revision has answered all the questions raised by the reviewer and met the standard of the publication for Scipost.

Recommendation

Publish (surpasses expectations and criteria for this Journal; among top 10%)

---

## Round 2 · Author Response

We have incorporated and made changes in the script as per the referee comments . We have also mentioned the places where we have made changes in the reply to the referee's comment. Please refer the link below.

https://scipost.org/submissions/2401.15948v1/

---

## Round 2 · List of Changes

1. In Section Introduction, page no. 4, paragraph 2, line no. 20, we have added the advantages of our method as well as reply to the first point raised in Report 2 by the referee.

2. In Appendix C, under Hyperparameter Details, we have addressed the third point raised in Report 2 .

3. Figs. 5 & 6 have been updated along with captions as referred to in points 5, 6, and 7 raised in Report 2.

4.Typo above Eq. 7 has been amended as raised in point 10 in Report 2.

5. We have added the results for 32x32 lattice size trained on 10 temperature settings in the revised manuscript in Table 8 and Fig.7 .

6. In Conclusion, Paragraph 4, Page no. 18, line no 12, we have discussed the second point raised in Report 1.

7. In Conclusion, last paragraph, Page-19, we have included the future work addressing scaling up to large lattices with factorizable model.

---

## Editorial Decision

accepted_in_target_journal